

# The interplay between cognitive tasks and vision for upright posture balance in adolescents

Hai-Jiang Meng[*], Shan-Shan Luo[*] and Yuan-Gang Wang

School of Sports, Anqing Normal University, Anqing, China
[*] These authors contributed equally to this work.

Corresponding author
Yuan-Gang Wang,
1371133055@qq.com

## ABSTRACT

**Background**. The control of an upright stance in humans is important in medicine, psychology, and physiology. The maintenance of upright stance balance depends not only on sensory information from proprioceptive, vestibular, cutaneous, and visual sources but also on cognitive resources. The present study investigated the effects of cognitive tasks while standing with eyes open on upright stance balance in adolescents. We hypothesized that performing a cognitive task while standing with eyes open would increase body sway among these adolescents and that the upright posture would thus become less stable.

**Methods**. A static balance assessment system comprising a force platform connected to a computer was used to evaluate the stability of the upright stance among 21 healthy adolescents under six conditions: no cognitive task, a relatively easy cognitive task, or the same cognitive task made more difficult, with each task being performed while the eyes were open and again while the eyes were closed. The participants performed mental calculations as fast as possible by subtracting either 3 or 18 from a random three-digit number continuously, for the simple cognitive task or the difficult cognitive task, respectively. Each calculation was completed within 10 s. The evaluation indexes used to measure upright posture stability were the root mean square (RMS) of the total body sway in the mediolateral and anteroposterior directions, the mean velocity (MV) value of the total body sway, and the Romberg quotient (RQ) of these values.

**Results**. The RMS ($p < 0.01$) and MV ($p < 0.01$) values of the upright posture sway were lower when participants performed no cognitive task and their eyes were open than when their eyes were closed. When their eyes were open, compared with no cognitive task, the values of the measures evaluating upright posture sway were higher, meaning the stance was less stable, while performing either the simple or the more difficult cognitive task (RMS: simple task, $p < 0.01$; difficult task, $p < 0.05$; MV: simple task, $p < 0.01$; difficult task, $p < 0.01$) although no significant differences were detected for the RMS or MV values between the simple and more difficult cognitive tasks. The RQs for both the RMS and the total MV values of the upright posture sway during performance of the difficult cognitive task were significantly lower than when the participants performed no task.

**Conclusion**. Performance of a cognitive task significantly reduced the upright posture balance in adolescents during eyes open although increased task difficulty did not show a greater effect. The interference between the performance of a cognitive task and the visual control of an upright stance may be attributable in part to cognitive and

visual processing streams competing for common central resources, consistent with the Multiple Resource Theory of information processing.

## INTRODUCTION

Upright walking is arguably considered uniquely human. Controlling posture stability is considered the premise and basis of other kinds of action behaviors. Thus, understanding posture control to maintain an upright stance among humans is an important issue in modern medicine, psychology, and physiology (*Ren, Watanabe & Miyatani, 2010*). Recent studies have concluded that balance maintenance in the upright body position is not entirely automated. It not only depends on sensory information from proprioceptive, vestibular, cutaneous, and visual sources (*Ceyte et al., 2014*; *Lakhani & Mansfield, 2015*; *Manor et al., 2010*), but also requires a large amount of cognitive resources (*Chong et al., 2010*; *Polskaia et al., 2015*). Some studies have found that when processing cognitive tasks in a constantly changing environment, upright posture control systems regulate the body's postural sway through complex interactions of somatosensory, visual, and vestibular sensory feedback networks with the musculoskeletal systems (*Manor et al., 2010*; *Muelas Perez et al., 2014*). Therefore, the modulation of cognitive tasks on the visual control of upright posture has attracted the attention of researchers.

The interaction between cognitive tasks and vision adjusts the stability of the upright posture. One study found that performing a cognitive task while standing with closed eyes increases body sway (*Donker et al., 2007*). According to the Multiple Resource Theory of information processing (*Wickens, 1980*), the interference between the performance of a cognitive task and the visual control of an upright stance may be attributable to cognitive and visual processing streams competing for common central resources (*Fraizer & Mitra, 2008*). Therefore, the effects of cognitive tasks on the stability of upright posture differ under visual and non-visual conditions. Recent evidence supports this theory, showing that performance of cognitive tasks inhibit visual processing during upright posture control (*Ren, Watanabe & Miyatani, 2010*). However, it remains unclear whether the difficulty of the cognitive task plays a key role, that is, whether there is a difference between the modulation that occurs during performance of a simple cognitive task or of a more difficult cognitive task, in the visual control of upright posture.

Measuring the ability to balance is an important index for evaluating body function, and having proper balance is of substantial significance for numerous actions, for example, to prevent injuries from falls and improve sports skills among people of all ages. In adolescents, proper balance is important to promote growth and development, in general and to develop mechanical skills, including those used in sports. To provide additional evidence in support of the information processing Multiple Resource Theory, the present study investigated the effects of varying the difficulty of a cognitive task while standing with

eyes open on the upright posture balance among adolescents. Because, according to the theory, competition for limited information processing resources will impair processing of specific sensory channels, we hypothesized that performing a cognitive task while standing with eyes open would increase body sway among these adolescents and that the upright posture would become less stable.

## MATERIALS AND METHODS

### Participants

In total, 21 healthy adolescents from the same high school (10 males and 11 females) aged 16–19 years were selected who met the study criteria. Their mean age was $17.81 \pm 1.12$ years old, their mean height was $167.4 \pm 6.52$ cm, and their mean weight was $58.43 \pm 10.06$ kg. All participants were healthy, with no brain, nerve, or muscle disorders and had normal or corrected-to-normal vision. The study protocol was approved by the ethics committee of Anqing Normal University (Ethical Application Ref: ANU201112). Participants, or their guardians for adolescents younger than 18 years, were informed of the experimental details and signed informed consents prior to beginning the study.

### Experimental design

The experiment used a within-subjects $3 \times 2$ factorial design. The first factor was the visual condition, and it was divided into two levels: eyes open and eyes closed. The second factor was performance of a cognitive task, and it was divided into three levels: no cognitive task, a simple cognitive task, and a difficult cognitive task.

The upright posture balance was evaluated using two indicators, namely, the root mean square (RMS) of the body sway in the mediolateral (ML) and anteroposterior (AP) directions and the mean velocity (MV) of the total body sway, as well as the Romberg quotient (RQ) of these two indicators. The RMS and MV values are the two most commonly used indicators for evaluating the stability of upright posture. The RQ was calculated as the ratio of the indicator values (RMS or MV) measured when the eyes were open versus when they were closed (*Njiokiktjien & Van Parys, 1976*). Higher RQ values indicate a greater role for vision in posture control.

### Experimental equipment and materials

The experimental equipment was a human static balance assessment system—a force platform connected to a computer, called a dynamometer—developed by the Hefei Institute of Intellectual Science of the Chinese Academy of Sciences. The system has a high-precision five-dimensional force sensor, high-precision amplifier circuit, and 16-bit A and D sampling cards. The system collects, processes, and analyzes changes in the center of the pressure of the static upright posture of the human body and presents the analyzed data as intuitive graphics and curves. A computer (Windows 7, CPU 3.0 GHZ, display 19 inches, resolution $1,280 \times 720$ pixels, refresh frequency 100 Hz) was connected to the dynamometer to control the experimental flow and to record experimental data online. The computer was also connected to the pressure-sensing devices to enable an integrated perception of the somatic balance and the real-time recording of the motion graphics.

## Experimental tasks

For the main task, the participants were required to stand quietly on the dynamometer, maintaining their balance in an upright posture with a minimum of body sway either with their eyes open or closed, as requested by the investigator. During the upright posture balance test, the participants were asked to stand as still as possible on the force platform with their feet together (one foot touching the other), with their arms relaxed by their sides, and to look at the visual target on the wall in front of them.

For the secondary task, participants were required to stand on the dynamometer and to maintain their balance in an upright posture with their eyes open or closed while performing the assigned cognitive task. The cognitive task comprised a relatively simple or more difficult mental calculation. At the beginning of the experiment, the investigator gave the participant a randomly selected three-digit number, such as 423. The participant subtracted either 3 (for the simple cognitive task) or 18 (for the more difficult cognitive task) from the three-digit number continuously as fast as possible. The participants performed the mental calculations and gave oral reports as quickly and accurately as possible within 10 s for each task. If the participant made an error, the correct value was immediately provided, and the participant continued the mental calculations using the corrected value until the end of the task. For each condition, if the number of reported errors during a cognitive task exceeded three, the participant was required to repeat the task.

To perform the visual and non-visual portions of the experiments, the participants stood in the laboratory 2 m from a white wall that had a target (a black circle with a diameter of 4 cm) drawn on the wall at approximately eye level. During the visual control component of the experiment, the participants stared at the target, and during the non-visual control component, the participants closed their eyes.

Before the formal experiment began, we conducted a pilot study with eight participants (none of whom participated in the formal experiment). The participants completed the two mental calculation tasks as described above. The results of a one-way analysis of variance (ANOVA) of the correct number of mental calculations in 10 s for each task condition indicated that the difference in difficulty between the two tasks was substantial and significant ($F_{(1,15)} = 269.73$, $p < 0.001$), indicating that the difficulty levels of the cognitive tasks met our experimental requirements.

## Experimental procedures

The experiment was conducted in a university physical fitness promotion experimental center. The laboratory was quiet, and the environment was comfortable. Before the experiment, all participants were informed of the task requirements.

For balance assessment of the upright posture, each participant prepared according to the instructions presented by the computer and began the formal experiments. For the eyes open or closed variable, all the participants completed the upright posture stability test under three conditions: no cognitive task, the simple cognitive task, and the difficult cognitive task. Thus, there were six conditions: (1) no cognitive task with eyes open, (2) no cognitive task with eyes closed, (3) simple cognitive task with eyes open, (4) simple cognitive task with eyes closed, (5) difficult cognitive task with eyes open, and (6) difficult cognitive

task with eyes closed. These six conditions were randomly presented to the participants. One balance trial was performed for each of the six conditions. The experimental time for each condition was 10 s, and the rest time between each test condition was 2 min. All experiments were conducted at approximately the same time each day (after 2:00 PM) to minimize the differences in individual biological rhythms.

## Statistical analysis

A two-way repeated measures ANOVA was conducted to evaluate the effects of cognitive task and visual control on the RMS and MV values of the upright posture movement trajectory. Paired $t$ tests with Bonferroni corrections for multiple comparisons were used for post hoc tests if the results of the ANOVA showed significant main effects or interactions. In addition, a one-way ANOVA was used to investigate the differences in RQ values when participants performed no cognitive task, the simple cognitive task, and the relatively more difficult cognitive task.

SPSS software, version 17.0, was used for all statistical analyses. The threshold for statistical significance was $\alpha < 0.05$. Values are represented herein as the mean $\pm$ standard error (SE).

# RESULTS

## RMS of upright posture sway in adolescents

Figure 1 shows the mean and SE of RMS values of upright posture sway for participants under the six experimental conditions. For upright posture sway in the ML direction, the repeated measures ANOVA results showed that the main effects of both the visual ($F_{(1,20)} = 9.86, p = 0.005$), and cognitive ($F_{(2,40)} = 17.01, p < 0.001$) factors were significant, as was their interaction ($F_{(2,40)} = 3.5, p = 0.04$). The post hoc analysis showed that the upright posture sway amplitude when participants performed no cognitive task was smaller with the eyes open than with the eyes closed ($p < 0.01$). When participants had their eyes open, their sway amplitude was significantly smaller when they performed no cognitive task than when they performed either the simple cognitive task ($p < 0.01$) or the more difficult cognitive task ($p < 0.05$); however, there was no significant difference detected in sway amplitude between the simple or the more difficult cognitive task. That is, increasing task difficulty did not further increase sway amplitude.

For the upright posture sway in the AP direction, although the main effect of the cognitive factor was significant ($F_{(2,40)} = 7.48, p = 0.002$), neither the main effect of the visual factor ($F_{(1,20)} = 1.55, p = 0.227$) nor the interaction between these two factors was significant ($F_{(2,40)} = 2.23, p = 0.143$).

## MV of upright posture sway in adolescents

Figure 2 shows the change in the MV values of upright posture total sway for the participants under the six experimental conditions. The repeated measures ANOVA results showed that the main effects of the cognitive ($F_{(2,40)} = 14.27, p < 0.001$) and visual ($F_{(1,20)} = 6.02, p = 0.023$) factors were significant, and their interaction was also significant ($F_{(2,40)} = 3.23, p = 0.045$). The post hoc test results indicated that the MV value was smaller with the eyes open

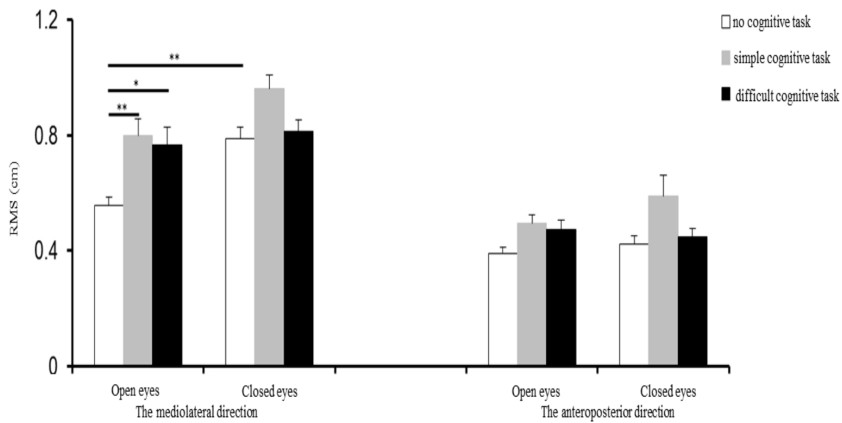

**Figure 1** **RMS of upright posture sway in adolescents under various experimental conditions.** $^*p < 0.05$; $^{**}p < 0.01$; RMS, root mean square.

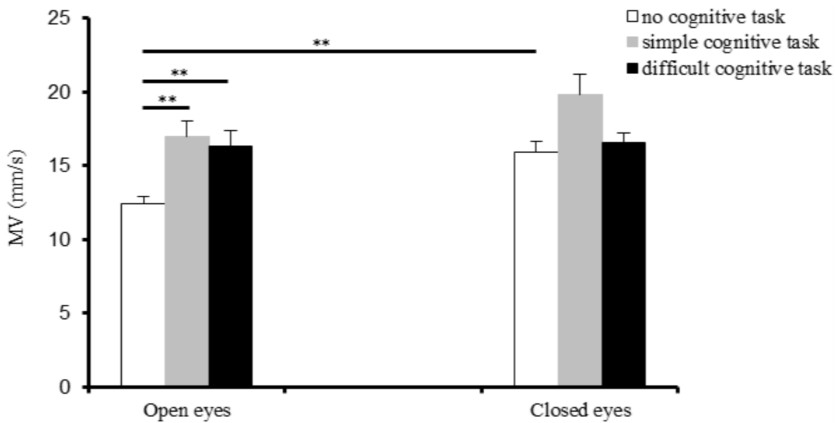

**Figure 2** **Mean Velocity (MV) of upright posture sway in adolescents under various experimental conditions.** $^{**}p < 0.01$.

than with the eyes closed while participants performed no cognitive task ($p < 0.01$). With the eyes open, the MV value of the upright posture sway was smaller when participants did not perform a cognitive task than when they performed the simple ($p < 0.01$) or the more difficult cognitive task ($p < 0.01$); however, there was no significant difference in the MV values of the upright posture sway between the simple and more difficult cognitive tasks.

## RQ of upright posture sway in adolescents

We compared the RQs for the RMS and the MV values for upright posture sway under the different task conditions (Fig. 3). The results showed that the RQ for the RMS value was significantly higher when participants performed no cognitive task than when they performed the more difficult cognitive task only for upright posture sway in the ML direction ($F_{(2,60)} = 3.63$, $p = 0.01$). The RQ for the total MV value was significantly higher
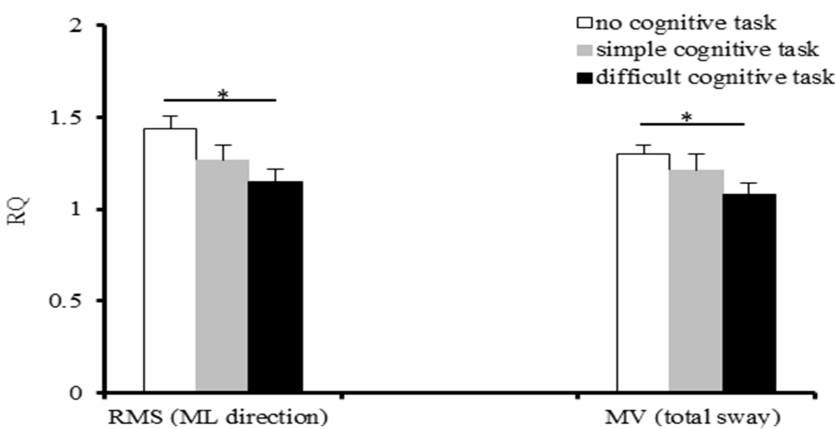

**Figure 3  Romberg Quotient (RQ) of upright posture sway in adolescents under various experimental conditions.** *p < 0.05.

when participants performed no cognitive task than when they performed the difficult cognitive task ($F_{(2,60)} = 2.81$, $p = 0.022$).

# DISCUSSION

## Visual control of upright posture balance

The RMS values for the ML direction and the total MV values for the upright posture sway showed that while adolescents performed no cognitive task, their upright stance was more stable when their eyes remained open than when their eyes were closed, indicating that vision affected the control of upright posture balance in these adolescents. Thus, the results of this portion of our study support previous findings on the importance of vision in upright posture control (*Lakhani & Mansfield, 2015*; *Muelas Perez et al., 2014*). In contrast to our results for the ML direction, the RMS values in the AP direction were not affected by our experimental manipulations. The front and rear of the feet are supported by the forefoot and the heel, respectively. Thus, the sway of an upright posture in the AP direction is not easily affected by the presence or absence of vision, and the upright posture remains relatively stable.

The control of upright posture balance depends on visual information. For example, studies have shown that when the visual field is limited or visual movement feedback is reduced, the amplitude of the upright posture sway increases, and conversely, when the visual field or movement feedback is no longer limited, the sway amplitude is reduced (*Hafström et al., 2002*; *Muelas Perez et al., 2014*). However, the maintenance of upright posture balance is a complex, multidimensional process, depending not only on visual information but also on other sensory information, such as proprioceptive, vestibular, and skin sensations (*Manor et al., 2010*). Indeed, other factors may also affect the processing of visual information for maintaining an upright stance and further research is needed to fully understand all of these factors.

## Effect of cognitive tasks on upright posture

The main effect of cognitive factor was found to be significant for open eyes in the analysis of RMS values for the ML and AP directions and for the MV of total body sway of the upright posture in adolescents. This result is consistent with those of some previous studies showing that cognitive processing plays an important role in upright posture control and supports the view that cognitive factors affect the balance control of the upright stance (*Lajoie et al., 2017*; *Polskaia et al., 2015*). However, the results of other studies are not consistent with those or with our results. For example, a few studies among young adults found that the upright posture sway increased (*Ceyte et al., 2014*; *Polskaia et al., 2015*), decreased (*Stins, Roerdink & Beek, 2011*), or was not altered (*Yardley et al., 1999*) in young adults during the execution of cognitive tasks; even when the complexity of the cognitive task was increased, their results remained unchanged. This apparent lack of consistent experimental evidence may be attributable to many factors, such as the processing needs of the specific cognitive task (*Fraizer & Mitra, 2008*), the difficulty of the postural task (*Barra et al., 2006*), and the use of reinforcement strategies (*Stins, Roerdink & Beek, 2011*). Thus, the performance of cognitive tasks may interact with other factors to influence upright posture balance.

## The interplay between cognitive tasks and vision for upright posture

Visual function, cognitive function, and stability control of the upright posture among humans are not independent systems. *Donker et al. (2007)* found that processing cognitive tasks while standing with closed eyes increases body sway. *Ren, Watanabe & Miyatani (2010)* examined the effects of cognitive tasks on the visual control of upright posture and found that performance of cognitive tasks inhibited visual processing for the control of upright posture. Similar to those two studies, our findings also supported the view that interactions between visual processing and cognitive task performance influenced upright posture control. Sway amplitude was larger during performance of a cognitive task than when no cognitive task was being performed when the eyes remained open, indicating that cognitive task processing reduces the role of vision in upright posture control. Because previous studies of maintaining upright posture control while performing cognitive tasks have indicated that the mental subtraction task increases body sway among the participants (*Donker et al., 2007*; *Ren, Watanabe & Miyatani, 2010*), we used both a relatively simple and a more difficult level of that task in the present study. The analysis of the RQ for the RMS values of the upright posture sway in the ML direction showed that the RQ with no cognitive task was higher than that for the more difficult cognitive task. These results further demonstrated that the role of vision in upright posture control was reduced during cognitive processing.

A current theoretical explanation of the control of upright posture balance is the Multiple Resources Theory of information processing; this theory holds that the interference between cognitive task performance and upright posture control is attributable to the competition for a common central pool of processing resources (*Ren, Watanabe & Miyatani, 2010*; *Riley et al., 2012*). On the basis of this theory, we speculated that cognitive task performance would affect visual control of upright posture balance. Our results showed that when the

participants' eyes were open, the upright posture sway while performing either the simple or the more difficult cognitive task was greater than that while not performing a cognitive task. Thus, our results support the Multiple Resources Theory of information processing. Owing to limited information processing resources, the ability to maintain upright posture balance will compete with cognitive tasks and with vision for central processing resources, resulting in an increase in the amplitude of the upright posture sway.

Upright posture balance has been used to evaluate physical function among adolescents; indeed it has become an important indicator in the monitoring of physical fitness in adolescents. However, the use of this measure as indicator is not ideal. The maintenance of balance in an upright stance or during walking is a complex and multidimensional process requiring higher levels of motor control and cognitive flexibility to cope with balance threats (*Buchman et al., 2011*). Therefore, it may be too simplistic to suggest that determining the ability of an adolescent to balance in an upright stance is a good measure of their physical fitness without comprehensively evaluating their stability under the interference of other factors, such as their simultaneous performance of cognitive tasks. Furthermore, for balance training of young people, the addition of interfering factors should be considered to improve the training effectiveness.

In summary, while processing cognitive tasks, the role of vision in the upright posture control of adolescents is significantly reduced, which indicates that visual function, cognitive function, and human upright posture stability control are not independent systems. These findings further support the view that upright posture control requires multisensory channel processing.

## CONCLUSIONS

The performance of a cognitive task while standing with eyes open reduced the upright posture balance in adolescents although this posture control was not further reduced by the simultaneous processing of a more difficult cognitive task. The interference between cognitive task performance and visual control of stance may be attributable in part to the competition between cognitive and visual processing for common and limited central processing resources.

### Funding

This work was supported by the Philosophy and Social Science Planning Project of Anhui province (No. AHSKY2018D58), the Youth Research Fund Project of Anqing Normal University (No. SK201112). The funders had a role in study design, data collection and analysis, decision to publish, or preparation of the manuscript.

### Grant Disclosures

The following grant information was disclosed by the authors:
Philosophy and Social Science Planning Project of Anhui province: AHSKY2018D58.
Youth Research Fund Project of Anqing Normal University: SK201112.

## Competing Interests

The authors declare there are no competing interests.

## Author Contributions

- Hai-jiang Meng conceived and designed the experiments, performed the experiments, contributed reagents/materials/analysis tools, prepared figures and/or tables, authored or reviewed drafts of the paper, approved the final draft.
- Shan-Shan Luo conceived and designed the experiments, performed the experiments, analyzed the data, contributed reagents/materials/analysis tools, prepared figures and/or tables.
- Yuan-Gang Wang conceived and designed the experiments, analyzed the data, authored or reviewed drafts of the paper, approved the final draft.

## Human Ethics

The following information was supplied relating to ethical approvals (i.e., approving body and any reference numbers):

The study protocol was approved by the ethics committee of Anqing Normal University (Ethical Application Ref: ANU201112).

## Data Availability

The raw measurements are available as Supplemental File.

## Supplemental Information

Supplemental information for this article can be found online at http://dx.doi.org/10.7717/peerj.7693#supplemental-information.

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
