# Peer review of "The interplay between cognitive tasks and vision for upright posture balance in adolescents"

_PeerJ, doi:10.7717/peerj.7693_

## Round 0.1 · original submission · Major Revisions

I invite the authors to carefully address the points indicated by the two reviewers and resubmit the paper indicating all the changes made to the manuscript and providing a point by point reply.

[]

·

Basic reporting

- It is strongly recommended to do an orthographic and grammatical review before it is printed. This is necessary not only to improve the quality of the manuscript, but also to avoid misinterpretations.
- The introduction should explain some topics in more detail (I have made several recommendations below, see the PDF-Data). The final part of the introduction should be restructured in order to present a more logical and clear sequence of ideas until the hypothesis is reached.
-Figure 2 and 3 should provide more information (see details in PDF)

Experimental design

- The research question is definitely an issue of relevance. However, in many situations -probably due to language limitation- it is not possible to identify important methodological issues, such as the hypothesis of the study.
- Some aspects about the methods were asked (see PDF attachment) in order to provide sufficient details and information for a future replication.

Validity of the findings

- The choice of the statistical tests is correct. However, regarding the Bonferroni correction, some information needs to be offered (see PDF attachment)
- Regarding the interpretation of the result, I believe it is correct, but it has been extrapolated. I made some comments about in order to help you and avoid problems with wrong comprehension.

Additional comments

The authors investigated the influence of cognitive tasks on balance control (quasi-static balance) in adolescents, when performing with eyes open and eyes closed. This is definitely an issue of relevance, since there are controversies in the literature about dual task balance tests performed during eyes-open and eyes-closed conditions. Despite presenting numerous methodological and language problems, the study has good potential. In this way, I would like to contribute with some suggestions.

Dear Authors,
I have read your manuscript and generally, the study was sufficiently adequate for publication taking into consideration aspects like experimental design and validity of the findings. However, I have made several recommendations in order to make this study clearer and to improve its quality. Furthermore, there are some questions I would like to see answered related to the study. I would also like to recommend the reorganization of some information according to each section (I have made several recommendations below for each specific section). Additionally, it is also strongly recommended to do an orthographic and grammatical review before it is printed. I have also noted some sentences or phrases that should be rewritten. This manuscript is short and to the point but I believe the discussion could be better structured.

Reviewer 2 ·

Basic reporting

1.1. The authors used a clear English language.
1.2. Raw data was supplied.
1.3. Figures have a good quality and are relevant.
1.4. The authors presented the context related to balance maintenance and the role of cognition during balance tasks. The study was focused on adolescents individuals, however the context described above applied to this population and the gap existent in the literature are not well addressed and need to be improved.

Experimental design

2.1. The research question is well defined and relevant.
2.2. Some aspects should be improved and clarified in the Methods section:
- Did the adolescents included in the study present a similar cognition condition?
- What was the range of age of the included subjects?
- Was the number of errors controlled during the cognitive tasks? If yes, they need to be reported.
- It is not clear in the Methods section which was the total duration of each trial for the balance tasks, nor if the participants needed to do more than one subtraction for the same trial. Please, describe the total duration of each trial, the number of trials performed for each task and if the random 3-digit number was other for each trial.
- Please, describe the Romberg Quotient in the Methods section and for which direction it was determined. Maybe the information presented in the 3.3 section could be replaced in the Methods section.
- Another aspect that can influence the individuals’ balance is the base of support. Due to this, it is necessary to clarify what did the authors mean with the term “feet together” for the balance tests and how this position was controlled for all subjects (e.g. was there a distance between the feet? Or was one foot touching the other?).
- Please, specify in the Methods section and in the Abstract that the RMS will be reported for both ML and AP directions.

Validity of the findings

3.1. The main objective of the study was to investigate the effects of varying the difficulty of a cognitive task on the visual control of upright posture among adolescents. However, from the results presented, it is not clear if the difficult cognitive task in comparison to the simple cognitive task lead to alterations in the balance. It was only demonstrated that both were different from the no cognitive task, but the results for the comparison between them was not presented. Please, clarify this for all variables.
3.2. For the eyes closed tasks it is also necessary to demonstrate if there were differences between the tasks or if they were similar. From the figures presented, one can suppose that they were similar. Anyway, this need to be clearly stated for the readers.

Additional comments

1. The results presented in the Abstract are a little bit confused. Please, make clear for the readers what were the results according to the aim of the study.
2. In the Abstract section the term “difficultly” seems not appropriate. Did you mean “difficulty”?
3. The statistical analysis presented in the Results subsection in the Abstract should be replaced to the Methods subsection in the Abstract.
4. From the discussion and the title of the manuscript, one could think that when the difficulty of the task increase (no cognitive task, simple cognitive task and difficult cognitive task), the balance alters. However, from the results presented (as mentioned in the above comments) it is not clear if the difficult task was different from the simple cognitive task.

---

## Round 0.2 · Major Revisions

Dear authors

I invite you to carefully address the remaining comments from the reviewers which are described below and resubmit your paper.

·

Basic reporting

The authors changed significantly the manuscript, and improved its quality, especially regarding the language review.

Experimental design

Line 134: include that guardians signed the consent, when testing subjects younger than 18.

Validity of the findings

Please check the general comments about the interpretation of the data.

Additional comments

First, I want to congratulate the authors for the revision. I have still some concern about the interpretation of your result and, consequently, about the title of your paper. The generalization of the results is also an issue to revise.

- Line: 282-284: “The main effect of cognitive factor was found to be significant in the analysis of RMS values for the ML and AP directions and for the MV of total body sway of the upright posture in adolescents.” Please, specify that the results were found for open eyes.
- Line: 300-304: “Donker et al. (2007) found that processing cognitive tasks while standing with closed eyes increases body sway. ….. Our findings are consistent with those of both of these studies”
Please, check this again, because your results did not show significant increases in balance parameters for dual task conditions when eyes were closed.

- Line 322-327: especially “will lead to a reduction in the visual processing channels”. I would not affirm this/ or I would write this carefully, because you did not directly measure the flow of signals.
Another point about this affirmation is that: If the instability happened during dual tasks is explained by the fact that a “reduction of visual processing channels” occurs, it raised the question:
Why did you not find an effect of dual tasks in a situation in which the visual processing channels are naturally reduced, such as your eyes closed condition? I mean, your data showed no differences between both cognitive task conditions for any parameters when participant´s eyes were closed (EC).
Your data indeed demonstrates that the dual task has an effect on postural control during eyes open. However, you cannot affirm that the reason of this is based on the reduction of visual processing channels. As you also mentioned in your paper, postural control is a complex process, involving the integration of afferent information provided by various systems. The integration and the priority of afferent information from one or more systems seems to change constantly according to necessity, thus recruiting more information and reweighting the importance of each available input. Therefore, I think that it can be a speculation to state that a reduction of visual information occurs during dual tasks conditions and that it is the explanation for the instability in general. This is also because when the visual information was inhibited (eyes closed) in your study, you did not present effects of dual task in balance. Hence, I would suggest more caution in the interpretation/extrapolation of the data, since your affirmation is in the abstract, in the discussion, in the conclusion and also in the title of the manuscript.

Reviewer 2 ·

Basic reporting

No comments.

Experimental design

No comments.

Validity of the findings

No comments.

Additional comments

I would like to thank the authors for the paper revision. However, I have still some minor suggestions.
- It is necessary to clarify and contextualize in the Introduction section why this project is important to be performed with adolescents individuals. By this way, the importance of your study will be strengthened for the readers.
- Please, include in the Methods section how many balance trials were performed for each of the six conditions.
- It would be more appropriated to replace the second paragraph of the “Experimental procedures” subsection (Methods section) close to the first paragraph of the “Experimental tasks” subsection (Methods section), once that one completes the other.

---

## Round 0.3 · Minor Revisions

Dear Authors

Please take into consideration the comments from reviewer 1. Since you addressed properly the request, the paper will not go for review again, and I will take the editorial decision.

·

Basic reporting

The requested changes have been submitted.

Experimental design

The requested changes have been made.

Validity of the findings

The authors changed the text in the discussion “Owing to limited information processing resources…”. However, the interpretation/extrapolation of the data is still present in the abstract, in the conclusion, and in the title of the paper. Since you cannot affirm that a reduction of visual control happened, you should change this affirmation in the abstract, in the conclusion, and also in the title of the manuscript.

Additional comments

The authors answered the questions and made most of the requested changes. However, there are still some essencial changes to be made.

Reviewer 2 ·

Basic reporting

No comments.

Experimental design

No comments.

Validity of the findings

No comments.

---

## Round 0.4 · accepted · Accept

We acknowledge the effort of the authors to attend the comments from different reviews and congratulate the author for the acceptance of the paper for publication.

#